# Dextromethorphan Exhibits Anti-Inflammatory and Immunomodulatory Effects in a Murine Model: Therapeutic Implication in Psoriasis

**DOI:** 10.3390/life12050696

**Published:** 2022-05-07

**Authors:** Yi-Ming Chen, I-Chieh Chen, Ya-Hsuan Chao, Hsin-Hua Chen, Po-Ku Chen, Shih-Hsin Chang, Kai-Jieh Yeo, Shiow-Jiuan Wey, Chi-Chien Lin, Der-Yuan Chen

**Affiliations:** 1Division of Translational Medicine, Department of Medical Research, Taichung Veterans General Hospital, Taichung 40705, Taiwan; ymchen1@vghtc.gov.tw (Y.-M.C.); icchen@vghtc.gov.tw (I.-C.C.); 2Division of Allergy, Immunology and Rheumatology, Taichung Veterans General Hospital, Taichung 40705, Taiwan; shc5555@vghtc.gov.tw; 3Institute of Biomedical Science, Rong-Hsing Research Center for Translational Medicine, National Chung Hsing University, Taichung 402, Taiwan; 4Department of Post-Baccalaureate Medicine, College of Medicine, National Chung Hsing University, Taichung 402, Taiwan; 5School of Medicine, National Yang Ming Chiao Tung University, Taipei 11221, Taiwan; 6Program in Translational Medicine, The iEGG and Animal Biotechnology Center, National Chung Hsing University, Taichung 402, Taiwan; demonsandy@gmail.com (Y.-H.C.); sherry61976@hotmail.com (S.-H.C.); 7Division of General Medicine, Department of Medicine, Taichung Veterans General Hospital, Taichung, 40705, Taiwan; 8Department of Industrial Engineering and Enterprise Information, Tunghai University, Taichung 407224, Taiwan; 9Institute of Medicine, Chung Shan Medical University, Taichung 40201, Taiwan; 10Rheumatology and Immunology Center, China Medical University Hospital, Taichung 404332, Taiwan; pago99999@gmail.com (P.-K.C.); D30870@mail.cmuh.org.tw (K.-J.Y.); 11College of Medicine, China Medical University, Taichung 404332, Taiwan; 12Division of Dermatology, Chung Shan Medical University Hospital, Taichung 40201, Taiwan; weyth@tcts1.seed.net.tw; 13Department of Medical Research, China Medical University Hospital, Taichung 404332, Taiwan; 14Department of Pharmacology, College of Medicine, Kaohsiung Medical University, Kaohsiung 807, Taiwan

**Keywords:** psoriasis, dextromethorphan (DXM), immune regulation, T cell receptor γδ T cell, IL-17, IL-22

## Abstract

Psoriasis is an immune-mediated skin disease with a worldwide prevalence of 2–4% that causes scaling erythematous skin lesions. It is a chronic relapsing and complex multifactorial disease that often necessitates long-term therapy. Despite various novel therapies, psoriasis remains a treatable but non-curable disease. Because the antitussive medication dextromethorphan (DXM) can inhibit murine bone marrow and human monocytes and slow the progression of arthritis in mice with type II collagen-induced arthritis, we explored whether the oral administration of DXM to mice with imiquimod (IMQ)-induced psoriasis can effectively alleviate psoriasis symptoms and improve immune regulation. Herein, we examined the therapeutic effects of DXM on psoriasis and its potential mechanisms of action in an IMQ-induced psoriasis mice model. We found that an oral dose of DXM (10 mg/kg) could more significantly reduce psoriasis symptoms compared with intraperitoneal injection. Seven days after the oral administration of DXM, the Psoriasis Area and Severity Index (PASI) score was significantly decreased compared with that in the vehicle group. Furthermore, DXM treatment also significantly ameliorated the psoriasis symptoms and the histopathological features of psoriasis, including stratum corneum thickening, desquamation, and immune cell infiltration. Additionally, DXM reduced the mRNA levels of the cytokines TNF-α, IL-6, IL-17A, and IL-22 in skin and the percentage of IL-17A and IL-22 producing T cell receptor γδ T cells (TCRγδT). Taken together, our research demonstrated that DXM could inhibit keratinocyte proliferation and alleviate psoriasis symptoms, which suggests the potential application of DXM in the treatment of chronic inflammation and autoimmune diseases.

## 1. Introduction

Psoriasis is a chronic immune-mediated skin disease that affects 2–4% of the world’s population [1,2]. The pathophysiology of psoriasis involves epidermal thickening, keratinocyte hyperproliferation, and parakeratosis in the epidermis, as well as massive neutrophil infiltration in the dermis [3]. Psoriasis patients present with scaly patches, erythematous, and plaques that can affect anybody part where they are found [4]. Psoriasis and its comorbidities, including psoriatic arthritis, cardiovascular diseases, and mental depressive illness [5,6,7], can lead to a significant decrement in quality of life and can incur substantial medical expenses. Although the pathological mechanisms of psoriasis are not fully understood, a number of risk factors that have significant impacts on the occurrence of psoriasis are recognized, including genetic factors, environmental factors [8], lipid metabolism, angiogenesis, and inflammation [9].

The clinical manifestations of psoriasis reflect the sustained inflammation in the epidermis, with uncontrolled hyperproliferation and dysfunctional differentiation of keratinocytes. Approximately 90% of psoriasis cases are related to chronic plaque-type psoriasis [10]. The reason for psoriasis may be ascribed to the development and sustainment of psoriatic inflammation caused by disturbances in the innate and adaptive cutaneous immune responses [11,12]. Accumulative evidence indicates that regulatory T cells (Tregs) play a key role in psoriasis pathogenesis and that Treg dysfunction causes an aberrant release of proinflammatory cytokines, including tumor necrosis factor (TNF)-α, IL-6, IL-17, and IL-23, and activates NF-κB signaling [13,14]. NF-κB signaling changes the function of keratinocytes by affecting cell proliferation, differentiation, separation, and apoptosis [15,16]. Several immunological studies demonstrated that patients with psoriasis had abundant memory T lymphocytes that did not undergo natural apoptosis, including CD4^+^ and CD8^+^ T lymphocytes [17,18,19,20]. Most of the T lymphocytes of psoriasis patients have CD45RO cell surface markers. The fundamental therapies for psoriasis, which often necessitate long-term therapy, include topical therapies, phototherapies, conventional synthetic disease-modifying drugs (csDMARDs), and biological DMARDs (bDMARDs), which may improve immune dysregulation. Mild psoriasis is mainly treated with topical agents, moderate psoriasis needs phototherapy in addition to topical therapy, and severe cases are treated with immunomodulatory drugs and combination therapy. Among these drugs, IL17 and IL23 inhibitors seem to have the most promising results [21,22].

Dextromethorphan (d-3-methoxy-17-methylmorphinan, DXM), an antagonist of the *N*-methyl-d-aspartate receptor, is broadly used as an antitussive drug and is structurally similar to morphine and naloxone. It has been considered to have possible applications for neurological and mental issues, such as depression, stroke, traumatic brain injury, and epilepsy [23]. This effect was mediated via its anti-inflammatory property by inhibiting proinflammatory cytokines such as TNF-α released from microglia in the brain [24]. Moreover, DXM has been reported to regulate the progression of atherosclerosis, a chronic inflammatory disease, by suppressing oxidative stress [25]. Meanwhile, our recent studies have shown that DXM can ameliorate the degree of arthritis in mice with type II collagen-induced arthritis [26]. Because inflammatory cytokines, i.e., TNF-α, IL-17, and IL-23, are involved in the pathogenesis of psoriasis, the high safety profile of DXM and its general immunomodulatory effects are advantages in its clinical application as a potential therapeutic treatment for psoriasis [27]. Based on these findings, we hypothesized that the oral administration of DXM to mice with imiquimod (IMQ)-induced psoriasis might effectively alleviate the symptoms of psoriasis and improve immune dysregulation through its effects on Tregs and T cell receptor γδ T cells (TCRγδT). In this pilot study, we aimed to evaluate whether DXM could alleviate the symptoms of psoriasis and inhibit keratinocyte proliferation in a murine model with psoriasis.

## 2. Materials and Methods

### 2.1. Animals

Female BALB/c mice (weights 25 ± 2 g) were provided by the National Laboratory Animal Center (NLAC), NARLabs, Taiwan. The mice were housed in standard vivarium cages with standard bedding, temperature (25 ± 2 °C), relative humidity (45–55%), and autoclaved water, maintained on a 12 h light/dark cycle, and kept under specific pathogen-free conditions. Female mice were anesthetized by intraperitoneal injection with 100 mg/kg ketamine and 10 mg/xylazine. The animal procedures performed in these experiments were approved by the Institutional Animal Care and Use Committee of National Chung Hsing University, Taiwan (approval protocol no. 105-118).

### 2.2. Imiquimod (IMQ)-Induced Psoriasis Murine Model

Eight-week-old mice were topically administered a dose of 62.5 mg of 5% imiquimod cream applied to a shaved area (3 cm × 2.5 cm) on their back for six consecutive days. Untreated naive mice were used as the normal group, and the IMQ group was given a topical administration of imiquimod cream to induce psoriasis. The DXM-treated groups were administered with DXM at 5, 10, 20 mg/kg via oral gavage or i.p. injection following 2 h of IMQ treatment for 6 days. As an antitussive, the usual clinical dose of DXM in adult humans is 60–120 mg/day. The dose used in mice is 5, 10, and 20 mg/kg, which is equivalent to a dose of 25 mg/60 kg, 50 mg/60 kg, and 100 mg/60 kg in an adult human, and is considerably lower than the toxic concentration (LD50 in rats = 350 mg/kg; IPCS INCHEM Database) [28].

### 2.3. Histological Analysis

Skin samples were fixed in 4% paraformaldehyde in PBS and embedded in paraffin. The sections (5 µm) were then processed for hematoxylin and eosin (H&E) staining for histological analysis. With hematoxylin staining, the nuclei appeared blue, and with eosin staining, the cytoplasm appeared pink. The stratum corneum desquamation, the thickening of epithelial tissue and the infiltration of immune cells in the dermis were analyzed under a light microscope (Olympus BX41, Tokyo, Japan).

### 2.4. Flow Cytometry Analysis

The harvested spleens were crushed through mesh strainers to obtain the spleen cells, which were subsequently cultured in wells containing RPMI medium and coated with 5 μg/mL of anti-CD3 mAb (clone 145-2C11, BD Pharmingen, San Diego, CA, USA) and 2 μg/mL of anti-CD28 mAb (PV-1, Biolegend, San Diego, CA, USA) for 18 h to stimulate T cells [29,30,31]. After 14 h of stimulation, BFA was added to the orifice plate, and then the cells were flushed out, transferred to a flow tube, and washed twice with 0.1% BSA. Using 1% BSA as the base, the anti-mouse γδT-FITC (BD Biosciences, Franklin Lakes, NJ, USA) or anti-γδT-PE (BD Biosciences) antibodies were prepared at a ratio of 1:1000. A total of 100 uL of antibody solution was added to each tube, and the cells were stained for 30 min on ice. Then, the cells were washed twice with 0.1% BSA. Then, 100 μL of IC fixation solution was added to each tube, and the cells were fixed on ice for 20 min. After 20 min, the cells were washed twice with 1× permeabilization solution, and 1× permeabilization solution was used as the base to prepare anti-mouse IL-17A-FITC (BioLegend) and IL-22-PE (eBioscience™, San Diego, CA, USA) antibodies at a ratio of 1:1000. We added 100 uL of antibody solution to each tube and proceeded at room temperature. After staining for 30 min, the cells were washed twice with 1× permeabilization solution. Finally, stained cells were fixed with 1% paraformaldehyde and detected on a BD Accuri™ C5 cytometer CAT. NO. 657,214. (BD Biosciences, San Jose, CA, USA) and analyzed with BD Accuri™ C6 software version 1.0264.21 software.

### 2.5. Psoriasis Area and Severity and IndexScore (PASI)

The severity degree of psoriasis in mice with IMQ-induced psoriasis on day 6 was evaluated with the DermNet New Zealand PASI score [28]. DemNet’s PASI score is a tool used to measure the severity and extent of psoriasis and comprises three major parameters: erythema (redness), desquamation (scaling), and induration (thickness). According to the clinical plaque signs, there were five levels to score (zero points, no symptoms; one point, mild symptoms; two points, moderate symptoms; three points, severe symptoms; and four points, very severe symptoms).

### 2.6. RT-PCR

The RNAspin Mini kit (GE Healthcare, Little Chalfont, Buckinghamshire, UK) was used to purify total RNA from the back skin (48 h) and then, using a Transcriptor First Strand cDNA Synthesis Kit (Roche Diagnostics GmbH, Mannheim, Germany), we made cDNA from the DNA. SYBR Green Master Mix (Roche Diagnostics Ltd., Lewes, UK) was used for quantitative real-time PCR, and the primers mentioned below were used:TNF-α, F: 5′-GGCTGCCCCGACTACGT-3′ and R: 5′-CTCCTGTGGTATGAGATAGCAAATC-3; IL-6, F:5′-TGCCATTGCACAACTCTTTTCT-3′ and R: 5′-TCGGAG GCTTAATTACACATGTTC-3; IL-17A, F: 5′-TTTTCAGCAAGGAATGTGGA-3′ and R: 5′-TTCATTGTGGAGGGCAGAC-3′;IL-22, F: 5′-GAAGGCTGAAGGAGACAGTGAAA-3′ and R: 5′-GTTCCCCAATCGCCTTGA-3′; hypoxanthine guanine phosphoribosyl transferase 1 (HPRT), F: 5′-GTTGGATAAGGCCAGACTTTGTTG-3′ and R: 5′-GATTCAACTTGCGCCATCTTAGGC-3′ in the ABI 7500 Fast Real-Time system (Applied Biosystems; Thermo Fisher Scientific, Inc., Waltham, MA, USA). To evaluate gene expression, real-time RT-PCR was performed on target genes using cDNA from skin tissues. The 2^−ΔΔCT^ method was used to determine the relative expression of target genes after normalization to HPRT.

### 2.7. Statistical Analysis

The results are presented as the mean value ± S.D. obtained from at least three independent experiments. The statistical significance was performed using one-way or two-way ANOVA to compare multiple treatments GraphPad Prism (version 8 for Windows; GraphPad Software, La Jolla, CA, USA) and a *p* value < 0.05 was considered statistically significant.

## 3. Results

### 3.1. Oral Dextromethorphan Alleviates Skin Lesions in IMQ-Induced Psoriasis-like Mice Erythema and Scaling

To evaluate the therapeutic effect of DXM, we utilized the IMQ-induced psoriasis mouse model. Compared with the normal group, after inducing by IMQ, manifestations that were similar to psoriasis appeared on the shaved dorsal skin, including erythema, scaling, thickness, and inflammation, in both the IMQ/vehicle and IMQ/DXM-treatment groups. As shown in Figure 1A, mice with IMQ-induced psoriasis were treated with 10 mg/kg DXM by oral and intraperitoneal injection. Then, 7 days after treatment with DXM, overall skin lesions were reduced compared with those in the IMQ group. We found that the symptoms of psoriasis were decreased significantly in the groups that were orally administered DXM compared to the intraperitoneal injection groups. To further determine whether orally administered DXM can affect systemic immune responses, we initially examined the spleen weights on day 7 after IMQ treatment. In this study, our data identified significant spleen enlargement following 7 days of IMQ treatment in mice. However, the average spleen weights of the mice in the IMQ-treated-group were reduced after these mice were 10 mg/kg DXM by oral and intraperitoneal injection. Likewise, the oral group was also superior to the i.p. injection group, although there was no statistical difference (Figure 1B).

Then, we assessed the degree of the psoriasis-like lesion after DXM treatment, according to the PASI score. Surprisingly, we found that the scaling and thickness index of mice who were orally administered DXM was more obvious than that of mice administered DXM by intraperitoneal injection (Figure 2). Therefore, the follow-up experiment was carried out by orally administering DXM.

Subsequently, to determine the dose of DXM, we orally administered 5 mg/kg, 10 mg/kg and 20 mg/kg DXM to mice with IMQ-induced psoriasis. As shown in Figure 3A, under the 20 mg/kg and 10 mg/kg DXM treatment, the backs of the mice were smoother and had significantly fewer psoriatic skin lesions than those under DXM treatment (5 mg/kg DXM). However, there was no significant difference between the 20 mg/kg and 10 mg/kg groups. The same results were found in our analysis of spleen weight, where both the DXM 20 mg/kg and 10 mg/kg groups had significant weight reductions relative to the IMQ group, whereas the 5 mg/kg group was significantly lower than the IMQ group.

We next evaluated the psoriasis-like lesions on the backs of the mice with IMQ-induced psoriasis using PASI scoring [32] and found that the scaling and thickness severity of psoriasis was significantly reduced in 10 and 20 mg/kg DXM-treated mice compared with mice treated with doses of 5 mg/kg DXM (Figure 4A–C). However, there was no statistically significant difference between DXM 10 mg/kg and 20 mg/kg. Therefore, we considered 10 mg/kg to be the appropriate dose and we also used this dose for subsequent mechanistic analysis.

### 3.2. DXM Decreased the Severity of IMQ-Induced Psoriasis form Dermatitis

Histological examinations were performed to evaluate the effects of DXM on alleviating the symptoms of psoriasis and inhibiting keratinocyte proliferation in the IMQ-induced mouse model. As shown in Figure 5, the IMQ group had severe inflammation, abnormal alteration in epithelial structure with stratum corneum thickening (C), and increased acanthosis and hyperkeratosis of the epidermis (E) in the skin of mice compared to the control group. After treatment with 10 mg/kg DXM, the psoriasis symptoms of desquamation, immune cell infiltration, and stratum corneum thickening (Figure 5A,B) were significantly reduced.

### 3.3. DXM Reduced Inflammatory Cytokines in IMQ-Treated Mice’s Skin Lesions

To further investigate the effect of DXM on IMQ-induced dermatitis in mice, we performed a quantitative RT-PCR assay 48 h after IMQ administration to determine the mRNA expression of these cytokines in lesion skin. As illustrated in Figure 6, the expression of all cytokines was significantly higher in the IMQ/-treated group than in the normal group. However, mice administered 10 mg/kg DXM demonstrated a significant drop in TNF-, IL-6, IL-17A, and IL-22 levels in their skin.

### 3.4. DXM Decreased the Frequency of IL-17/IL-22-Producing TCRγδT Cells in the Spleens of IMQ-Treated Mice

In the imiquimod induced psoriasis mice model, CD4+ and TCR γδ T cells are believed to be major sources of IL-17 and IL-22 [33,34]. We determined the expression levels of IL-17A and IL-22 in TCRγδT cells in the spleens of psoriasis-like mice after treatment with DXM using flow cytometry. As depicted in Figure 7, the percentages of IL-17A and IL-22 expressing TCRγδT cells in mice increased after IMQ induction but decreased after 10 mg/kg DXM treatment.

## 4. Discussion

A previous study reported that the xenotransplantation model is regarded as the closest to incorporating the complete genetic, phenotypic, and immunopathogenic processes of psoriasis through the repeated topical use of imiquimod in mice causing the influx of various immune cells and hyperplasia of the epidermis, generating preclinical mouse models of psoriasis [35,36]. Here, we utilized the same mouse model of IMQ-induced psoriasis and found that orally administered DXM significantly decreased the PASI scores, reduced epidermal hyperplasia and abnormal proliferation, and diminished the induration of the epidermis compared to the vehicle group, indicating that a high dose of DXM via oral administration ameliorates IMQ-induced psoriasis in mice. It is well known that DXM is a D-isomer of the codeine analog levorphanol and is generally utilized as an antitussive in cough medications. In contrast to codeine, DXM is devoid of analgesic properties, causes less gastrointestinal disturbance and drug dependence, and has been safely administered orally at 10–40 mg/kg in mice [25]. Prior studies have reported that DXM has a novel antioxidative effect of reducing lipopolysaccharide-induced damage to dopaminergic neurons in rat neuron/glial cultures by restraining microglial activation with the ensuing diminished microglial production of superoxide and inflammatory cytokines [37] and reducing the elevation of serum TNF-α levels [38]. Additionally, DXM has also been demonstrated to reduce oxidative stress and inhibit atherosclerosis in a mouse model [25]. In this study, DXM treatment at 5, 10, and 20 mg/kg for six days decreased histological skin lesions and reduced epidermal thickening in mice with IMQ-induced psoriasis in a dose-dependent manner. Because DXM has a potent antioxidant effect, it can reduce the expression of proinflammatory cytokines, such as IL-6 and TNF-α [26], and inhibit inflammatory changes in psoriasis.

Psoriasis is an autoimmune disease with a drastic Th17 pathway involvement in the pathogenesis [15]. Meanwhile, Th17 cells produce the effector cytokines IL-17A and IL-22, which might play a marked pathogenic role in skin aggravation of psoriatic plaques [39,40]. In this study, we demonstrated that DXM suppressed IL-17- and IL-22-producing γδ T cells in the spleen. Th17-associated inflammatory cytokines at lesional skin were also significantly reduced following DXM treatment. Our result coincided with the finding that IL-17A- and IL-22-producing T cells in the peripheral blood of psoriasis patients play a critical role in affecting dermal inflammation [41]. In consideration of the inadequate response to IL-17 inhibitors and potential fungal infection encountered in patients with psoriasis, our study provided evidence supporting DXM as a potential add-on therapeutic without known immunosuppressive side effects. Further clinical trials are needed to confirm the therapeutic efficacy of DXM in patients with psoriasis.

In this study, we demonstrated the therapeutic effects of orally administered DXM on psoriasis and its potential mechanisms of action in IMQ-induced psoriasis mice. An oral dose of DXM (10 mg/kg) effectively alleviated the symptoms of psoriasis and improved immune regulation. Although DXM treatments reduced epidermal thickness, they could not fully suppress the symptoms of psoriasis. Therefore, the treatment of psoriasis should use a combination therapy, such as methotrexate, vitamin D, glucocorticoid ointment, anti-IL-17 monoclonal antibody, or phototherapy. In future research, conducting an efficacy evaluation of combination therapy is important and necessary.

## 5. Conclusions

Our study demonstrated that the oral administration of 10 mg/kg DXM alleviated the symptoms of psoriasis and inhibited keratinocyte proliferation by decreasing proinflammatory cytokines and IL-17- and IL-22-producing γδ T cell production. Our studies provide a valuable insight into the therapeutic potential of DXM in the treatment of psoriasis.

## Figures and Tables

**Figure 1 life-12-00696-f001:**
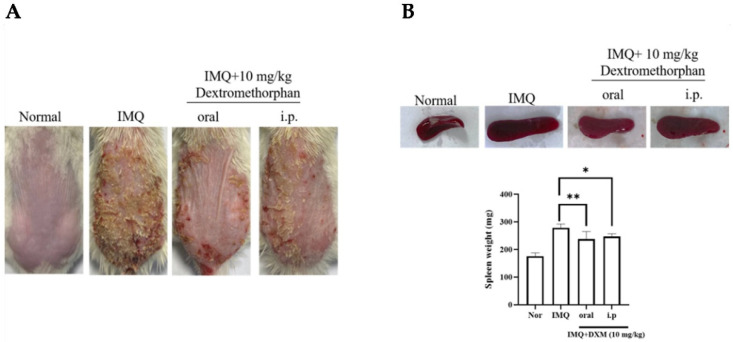
Oral DXM alleviates the clinical symptoms of IMQ-induced murine psoriasis. (**A**) The macroscopic appearance and the psoriasis area with 10 mg/kg treatment by oral and intraperitoneal injection. (**B**) The spleen weight of IMQ-induced psoriasis-like mice after DXM treatment. Significant differences from IMQ treated group are indicated by * *p* < 0.05, ** *p* < 0.01 (One-way ANOVA).

**Figure 2 life-12-00696-f002:**
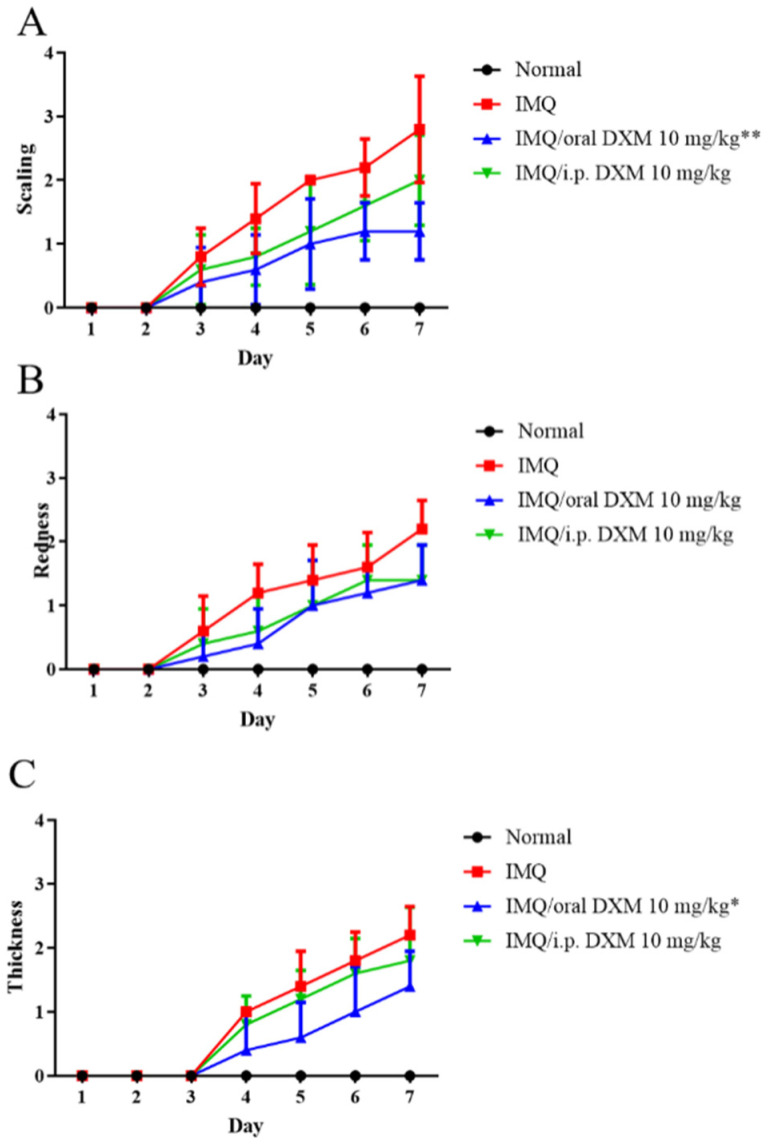
The subcomponents of psoriasis area and severity index (PASI) score, (**A**) Scaling, (**B**) Redness, and (**C**) Thickness, of the skin lesion were demonstrated in normal mice and IMQ-induced murine psoriasis (*n* = 5 in each group), after treatment with IMQ only (red), 10 mg/kg DXM by oral (blue) or intraperitoneal injection (green). * *p* < 0.05, ** *p* < 0.01 (Two-way ANOVA) versus the IMQ group.

**Figure 3 life-12-00696-f003:**
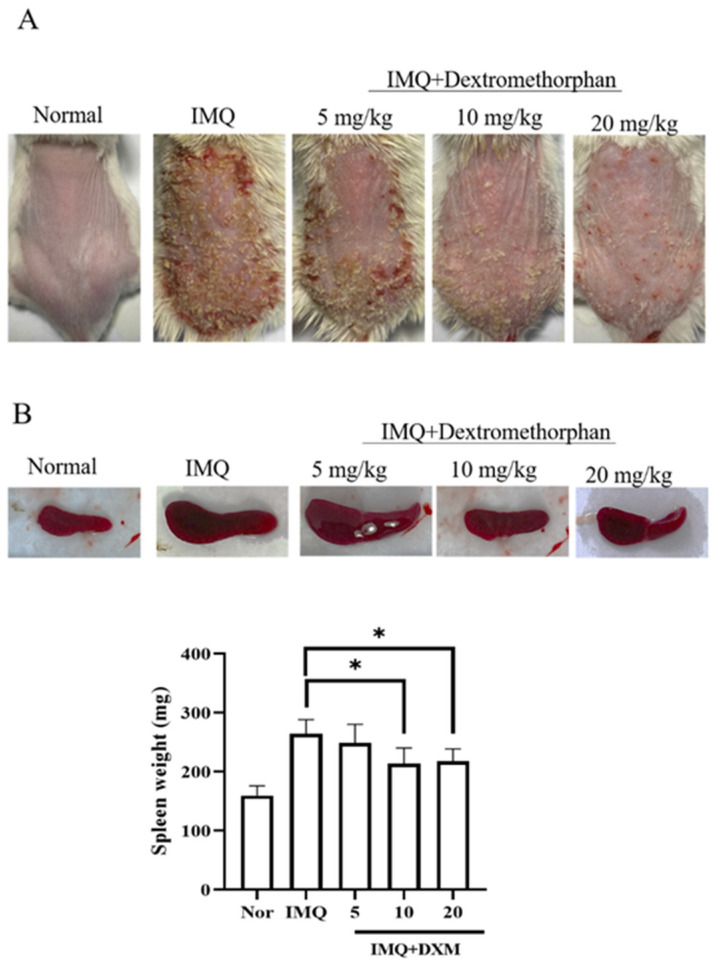
Clinical symptoms of psoriasis are regulated by various doses of DXM. (**A**) The macroscopic appearance and the psoriasis area with 5, 10 and 20 mg/kg DXM treatment by oral administration. (**B**) The spleen weight of IMQ-induced psoriasis-like mice after DXM treatment. Significant differences from IMQ treated group are indicated by * *p* < 0.05.

**Figure 4 life-12-00696-f004:**
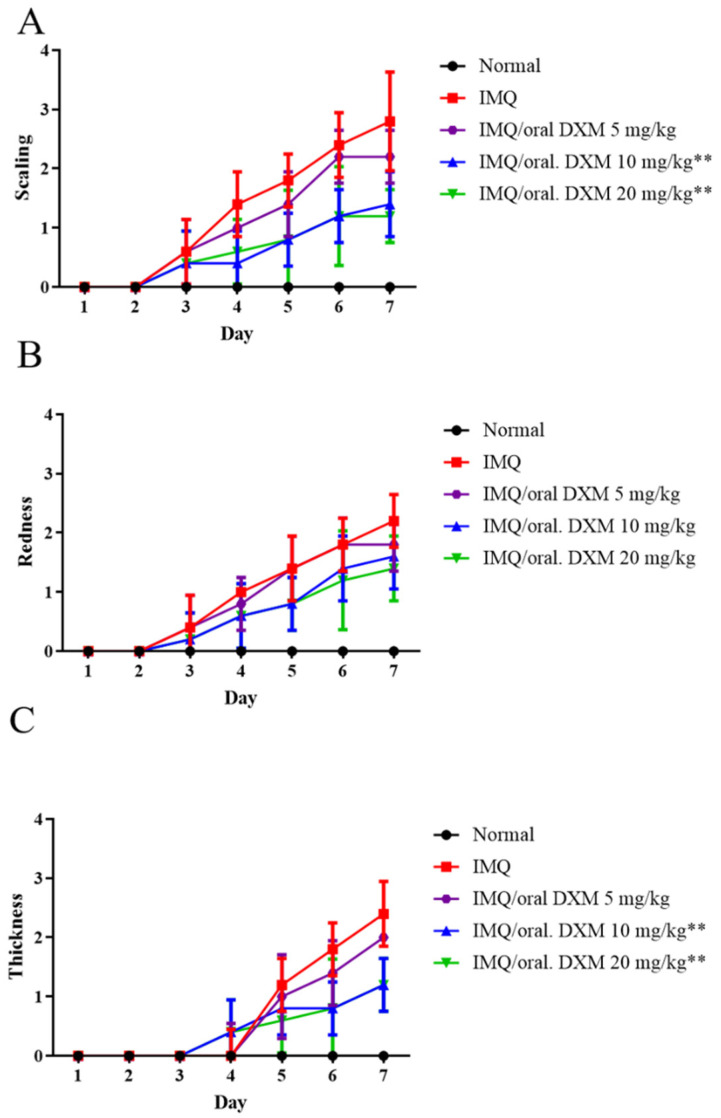
PASI score of the skin lesions in IMQ-induced murine psoriasis after treatment with 5, 10 and 20 mg/kg orally administered DXM. (**A**) Scaling, (**B**) Redness, and (**C**) Thickness. ** *p* < 0.01 (Two-way ANOVA) versus the IMQ group.

**Figure 5 life-12-00696-f005:**
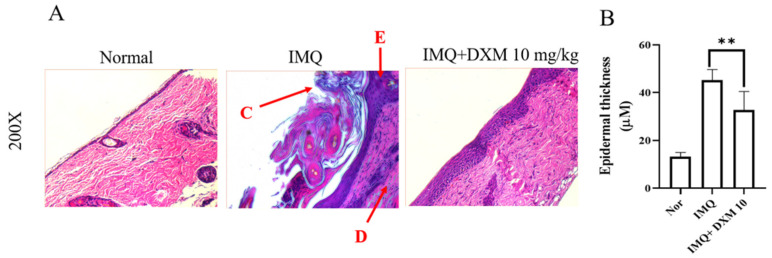
The effect of DXM on histological analysis of skin lesions of IMQ-induced psoriasis-like dermatitis in mice. (**A**) Representative H&E-stained back skin sections of different treatment groups. (C: Stratum corneum, E: Epidermis, D: Dermis) (**B**) Epidermal thickness of the dorsal skin on day 7. ** *p* < 0.01 (Two-way ANOVA) versus the IMQ group.

**Figure 6 life-12-00696-f006:**
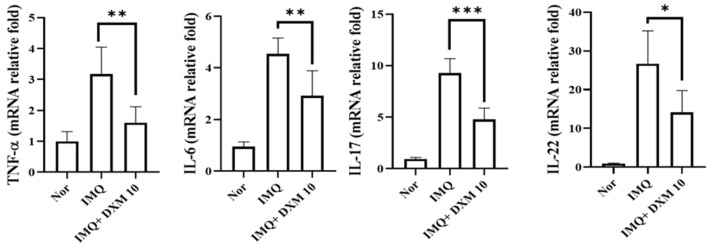
The effects of DXM on the generation of inflammatory cytokines in the skin of mice treated with IMQ. On day 7, skin was removed from separate treatment groups and RNA was extracted for quantitative RT-PCR analysis of TNF-, IL-6, IL-17A, and IL-22 mRNA expression. Internally, HPRT was used as a reference. The real-time PCR data are expressed as a fold change relative to the normal mouse group (set as 1.0). * *p* < 0.05, ** *p* < 0.01, *** *p* < 0.001 (One-Way ANOVA) versus the IMQ group.

**Figure 7 life-12-00696-f007:**
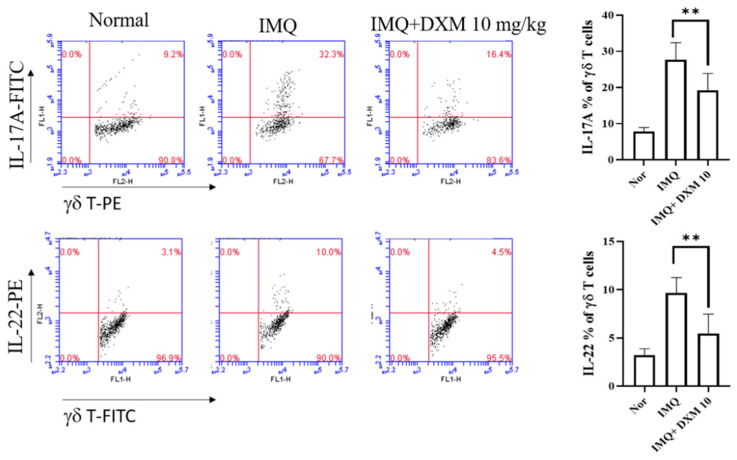
DXM decreased the percentage of IL-17- and IL-22-producing γδ T cells in the spleen. After seven days of treatment, the mice were killed, and their spleen cells were isolated. They were then incubated for 24 h with anti-CD3 and anti-CD28 flow cytometry was used to examine IL-17A and IL22 expression. Cells were gated for TCR γδ. The dot plot depicts data from an individual mouse from each group. The bar graphs show the mean SD for each group. ** *p* < 0.01, (One-way ANOVA) versus the IMQ group.

## Data Availability

The data that support the findings of this study are available from the corresponding author upon reasonable request.

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
