# Peer review of "Dextromethorphan Exhibits Anti-Inflammatory and Immunomodulatory Effects in a Murine Model: Therapeutic Implication in Psoriasis"

_life, 2022, doi:10.3390/life12050696_

Round 1

Reviewer 1 Report

An interesting original study demonstrating the therapeutic effects of orally administered dextromethorphan on psoriasis and its potential mechanisms of action on imiquimod-induced psoriasis in mice. 

This study may be a breakthrough in the management of psoriasis, as dextromethorphan may in the future be proposed for human studies; 

only minor queries:

line 81 you should add: "among these drugs, IL17 and IL23 inhibitors seem to have the most promising results" and cite: doi: 10.3390/healthcare9050543. and doi: 10.1111/dth.13170.

Thank You

Author Response

Author’s Response to reviewer’s comments

Reviewer #1

Comments and Suggestions for Authors

An interesting original study demonstrating the therapeutic effects of orally administered dextromethorphan on psoriasis and its potential mechanisms of action on imiquimod-induced psoriasis in mice. 

This study may be a breakthrough in the management of psoriasis, as dextromethorphan may in the future be proposed for human studies; 

only minor queries:

line 81 you should add: "among these drugs, IL17 and IL23 inhibitors seem to have the most promising results" and cite: doi: 10.3390/healthcare9050543. and doi: 10.1111/dth.13170.

Author's response:

Thank you for the constructive suggestion. We revised the manuscript and added these 2 references accordingly (lines 88-89).

Reviewer 2 Report

[Introduction]
-In the third paragraph, it seems necessary to add a reference to psoriasis and psoriatic arthritis after the sentence (Line 92-94) that the experiment was carried out based on the paper that dextromethorphan is effective for arthritis.

[Material and Methods]
-2.1 In Animals, it was said that mice (Five- to eight-week-old females) between 5 and 8 weeks of age were used, but there seems to be some margin of error between 5 and 8 weeks. And it is necessary to mention how many mice were used in each group.
-2.2. During the IMQ-induced psoriasis mouse model experiment, it is necessary to write in more detail whether IMQ was treated before or after Dextromethorphan.
-When confirming CD3 and CD28 by flow cytometry, the antibody detection time seems to be significantly longer than that of other papers (1-2h). Plaase explain why.

[Results]
-There are no A and B marks in Figure 1, and I am curious as to why the concentration was initially selected as 10mg/kg. (I wish there was a mention of previous experiments)
-It would be good if the explanations of each figure A, B, and C of Figure 2 were written in more detail.
-In the description of Figure 5 (Line 260-261), “severe inflammation” was mentioned, but it seems natural to mention it after adding related markers as IHC.
-When confirming the inflammatory cytokine in Figure 6, IFN-r, mentioned as an inflammatory cytokine in the Introduction and Discussion section, was not identified. Please add about the results.

[Discussion]
The part that mentions -mechanism (Line 312) is not clearly shown in the current figure, so it is better to delete it or add related data. Also, since the first paragraph is about Detromethorphan and the limitations of experimental results, it seems natural to move to the last part of the discussion.
-The oral treatment concentration of Line 339 needs to be revised to 5, 10, 20 mg/kg instead of 1, 5, 10

Author Response

Reviewer #2

Comments and Suggestions for Authors

[Introduction]
-In the third paragraph, it seems necessary to add a reference to psoriasis and psoriatic arthritis after the sentence (Line 92-94) that the experiment was carried out based on the paper that dextromethorphan is effective for arthritis.

Author's response:

Thank you for the constructive suggestion.

We added “Because inflammatory cytokines, ie. TNF-α, IL-17, IL-23, are involved in the pathogenesis of psoriasis, the high safety profile of DXM and its general immunomodulatory effects are advantages in its clinical application as a potential therapeutic treatment for psoriasis [27].” (lines 99-102)

Reference

  1. Singh, R.; Koppu, S.; Perche, P.O.; Feldman, S.R. The Cytokine Mediated Molecular Pathophysiology of Psoriasis and Its Clinical Implications. Int J Mol Sci 2021, 22, doi:10.3390/ijms222312793.

[Material and Methods]
-2.1 In Animals, it was said that mice (Five- to eight-week-old females) between 5 and 8 weeks of age were used, but there seems to be some margin of error between 5 and 8 weeks. And it is necessary to mention how many mice were used in each group.

Author's response:

Sorry for the misleading description. We purchased mice at about 5 weeks of age, and then conducted experiments at the age of 8 weeks. We also revised our description in the Material and Methods section 2.2. (line 110 & 120)

-2.2. During the IMQ-induced psoriasis mouse model experiment, it is necessary to write in more detail whether IMQ was treated before or after Dextromethorphan.

Author's response:

The DXM-treated groups were administered with DXM at 5, 10, 20 mg/kg via oral gavage or i.p injection following 2 hours of IMQ treatment for 6 days. (lines 123-125)

-When confirming CD3 and CD28 by flow cytometry, the antibody detection time seems to be significantly longer than that of other papers (1-2h). Please explain why.

Author's response:

We revised the misleading description. The purpose of our use of anti-CD3 and anti-CD28 is to stimulate T cell activation instead of staining. Previous studies indicated that naive CD4+ T cells require TCR stimulation for at least 18–24 h [29-31] for commitment to multiple rounds of cell division. We therefore chose 18 h as the stimulation time. (line 141-144)

Reference

  1. Schrum, A.G.; Palmer, E.; Turka, L.A. Distinct temporal programming of naive CD4+ T cells for cell division versus TCR-dependent death susceptibility by antigen-presenting macrophages. Eur J Immunol 2005, 35, 449-459, doi:10.1002/eji.200425635.
  2. Huppa, J.B.; Gleimer, M.; Sumen, C.; Davis, M.M. Continuous T cell receptor signaling required for synapse maintenance and full effector potential. Nat Immunol 2003, 4, 749-755, doi:10.1038/ni951.
  3. Iezzi, G.; Karjalainen, K.; Lanzavecchia, A. The duration of antigenic stimulation determines the fate of naive and effector T cells. Immunity 1998, 8, 89-95, doi:10.1016/s1074-7613(00)80461-6.

[Results]
-There are no A and B marks in Figure 1, and I am curious as to why the concentration was initially selected as 10mg/kg. (I wish there was a mention of previous experiments)

Author's response:

We added the A and B markers in Figure 1.

As an antitussive, the usual clinical dose of DXM in adult humans is 60–120 mg/day, and the estimated peak serum concentration is 8–16 μM30. The dose used in mice is 20 mg/kg, which is equivalent to a dose of 100 mg/60 kg in an adult human and is considerably lower than the toxic concentration (LD50 in rats = 350 mg/kg; IPCS INCHEM Database) [28]. (lines 125-129)

Reference

  1. Reagan-Shaw, S.; Nihal, M.; Ahmad, N. Dose translation from animal to human studies revisited. FASEB J 2008, 22, 659-661, doi:10.1096/fj.07-9574LSF.

-It would be good if the explanations of each figure A, B, and C of Figure 2 were written in more detail.

Author's response:

The figure legend of Figure 2 was revised as
The subcomponents of psoriasis area and severity index (PASI) score, (A) Scaling, (B) Redness, and (C) Thickness, of the skin lesion were demonstrated in normal mice and IMQ-induced murine psoriasis (n = 5), after treatment with IMQ only (red), 10 mg/kg DXM by oral (blue) or intraperitoneal injection (green). *p < 0.05, **p < 0.01 (Two-way ANOVA) versus the IMQ group. (lines 231-235)

-In the description of Figure 5 (Line 260-261), “severe inflammation” was mentioned, but it seems natural to mention it after adding related markers as IHC.

Author's response:

We revised this section as the following
As shown in Figure 5, the IMQ group had severe inflammation, abnormal alteration in epithelial structure with stratum corneum thickening (C), and increased acanthosis and hyperkeratosis of the epidermis (E) in the skin of mice compared to the control group. (lines 273-276)

-When confirming the inflammatory cytokine in Figure 6, IFN-r, mentioned as an inflammatory cytokine in the Introduction and Discussion section, was not identified. Please add about the results.

Author's response:

  1. In imiquimod induced psoriasis mice model, CD4+ and TCR γδ T cells are believed to be major sources of IL-17 and IL-22 [33,34]. Therefore, in this study, we mainly analyzed the degree of expression of IL-17 and IL-22 in γδ T cells. (lines 306-307)
  2. However, because the role of IFN-g in this animal model has not been demonstrated, and we did not analyze IFN-g-related immune responses in this study, we also removed IFN-g-related descriptions in the introduction and discussion sections.

Reference

  1. Van Belle, A.B.; de Heusch, M.; Lemaire, M.M.; Hendrickx, E.; Warnier, G.; Dunussi-Joannopoulos, K.; et al. IL-22 is required for imiquimod-induced psoriasiform skin inflammation in mice. J Immunol 2012, 188, 462-469, doi:10.4049/jimmunol.1102224.
  2. Cai, Y.; Shen, X.; Ding, C.; Qi, C.; Li, K.; Li, X.; et al. Pivotal role of dermal IL-17-producing gammadelta T cells in skin inflammation. Immunity 2011, 35, 596-610, doi:10.1016/j.immuni.2011.08.001.

[Discussion]
The part that mentions -mechanism (Line 312) is not clearly shown in the current figure, so it is better to delete it or add related data.

Author's response:

We agreed with the reviewer’s opinion and revised the manuscript. (lines 361-362).

Also, since the first paragraph is about Detromethorphan and the limitations of experimental results, it seems natural to move to the last part of the discussion.

Author's response:

We move the first paragraph of the discussion to the last part. (lines 358-365)

-The oral treatment concentration of Line 339 needs to be revised to 5, 10, 20 mg/kg instead of 1, 5, 10

Author's response:

We revised the concentration to 5, 10, 20 mg/kg, accordingly (line 341).